# Estimating Satellite and Receiver Differential Code Bias Using Relative GPS Network

**Alaa A. Elghazouly[1], Mohamed I. Doma[1], Ahmed A. Sedeek[2]**

[1] Faculty of Engineering, Menoufia University, Egypt.
[2] EL Behira Higher Institute of Engineering and Technology, El Behira, Egypt.

**Correspondence:** Alaa A. Elghazouly (alaa_elghazouly@sh-eng.menofia.edu.eg)

## Abstract

Precise Total Electron Content (TEC) are required to produce accurate spatial and temporal resolution of Global Ionosphere Maps (GIMs). Receivers and Satellites Differential Code Biases (DCBs) are one of the main error sources in estimating precise TEC from Global Positioning Systems (GPS) data. Recently, researchers are interested in developing models and algorithms to compute DCBs of receivers and satellites close to those computed from the Ionosphere Associated Analysis Centers (IAAC). Here we introduce a MATLAB code called Multi Station DCB Estimation (MSDCBE) to calculate satellites and receivers DCBs from GPS data. MSDCBE based on spherical harmonic function and geometry free combination of GPS carrier phase and pseudo-range code observations and weighted least square were applied to solve observation equations, to improve estimation of DCBs values. There are many factors affecting estimated value of DCBs. The first one is the observations weighting function which depending on the satellite elevation angle. The second factor concerned with estimating DCBs using single GPS Station Precise Point Positioning (PPP) or using GPS network. The third factor is the number of GPS receivers in the network. Results from MSDCBE were evaluated and compared with data from IAAC and other codes like M_DCB and ZDDCBE. The results of weighted (MSDCBE) least square shows an improvement for estimated DCBs, where mean differences from Center for Orbit Determination in Europe (CODE) (University of Bern, Switzerland) less than 0.746 ns. DCBs estimated from GPS network shows a good agreement with IAAC than DCBs estimated from PPP where the mean differences are less than 0.1477 ns and 1.1866 ns, respectively. The mean differences of computed DCBs improved by increasing number of GPS stations in the network.

**Keywords**: DCBs, Multi station, elevation angle, number of stations.

## 1. Introduction

TEC is an important parameter in the study of ionospheric dynamics, structures, and variabilities. The ionosphere is a dispersive medium for space geodetic techniques operating in the microwave band (Böhm, and Schuh, 2013) that allows calculation of TEC using GPS dual-frequency radio transmissions. The global availability of GPS has made it a valuable tool for sensing the Earth' the regional and global ionosphere estimation (Hernández-Pajares et al. 1999; Komjathy et al. 2005; Li et al. 2015; Liu and Gao 2004; Mannucci et al. 1993). Unfortunately, GPS-derived TEC measurements are adversely affected by an inherent interfrequency bias within the receiver and satellite hardware, typically referred to as the DCBs. Careful estimation of the DCBs is required to obtain accurate TEC, which is used in several applications, such as in several ionospheric prediction models, and in the correction of GPS positioning measurements (McCaffrey et al., 2017). A number of methods have been proposed for the estimation of GPS receiver DCBs, each with varying requirements and limitations including: making assumptions about the ionospheric structure; the use of internal calibration (Arikan et al., 2008; Themens et al., 2013,2015); or the use of a reference instrument or model. Estimating DCBs for receivers and satellites from GPS observations depending on two approaches, the relative and absolute methods. The relative method utilizes a GPS network, while the absolute method determines DCBs from a single station (Sedeek et al., 2017). In the current study, we applied relative method to calculate DCBs of satellites and GPS receivers.

There has also been growing interest in measuring the accuracy of these methods, and how different factors, e.g. ionospheric activity, plays a role in these methods (McCaffrey et al., 2017). Nowadays, reliable GIMs and accurate DCBs of satellites and The International GNSS Service (IGS) stations can be obtained from IAAC like CODE (Schaer,1999), European Space Agency (ESA, Germany) (Feltens and Schaer, 1998), Jet Propulsion Laboratory (JPL, USA) (Mannucci et al., 1998), and UPC (Technical University of Catalonia, Spain) (Hernández-Pajaresetal.,1999; Orús et al.,2005). However, the availability of IAAC DCB receivers' values, it is only available for IGS stations. Furthermore, some of IGS ground receiver DCB estimates are not available from all analysis centers. Also, some regions don't have any IGS ground stations like our country Egypt, which mean the TEC values over them would be interpolated from nearest calculated values. As TEC values depended on DCB values it is required a mathematical model to calculate DCBs from GPS data.

In this study we introduce a mathematical model estimating satellites & receiver DCBs for A GPS network based on Spherical Harmonic Function (SHF) written under MATLAB environment, the developed mathematical model uses geometry free combination of pseudo-range observables (P-code). Weighted Least Square was used to consider variation of satellites elevation angle. The code was evaluated and compared with other researchers' codes in section "Results and analysis". In the "Conclusion" section we summarize the overall paper results.

## 2. GPS Observation Model

For a GPS satellite, the pseudorange and carrier phase observations between a receiver and a satellite can be expressed as (Jin et al., 2008; Leandro, 2009; Leick et al., 2015; Zhang et al., 2018):

$$P_{r,j}^s(i) = \rho_r^s(i) + c(dt_r - dt^s) + T_r^s + I_{r,j,P}^s + DCB_r^P - DCB_s^P + M_j + E_j \qquad (1)$$

$$\Phi^s_{r,j}(i) = \rho^s_r(i) + c(dt_r - dt^s) + T^s_r - I^s_{r,j,\Phi} + \lambda_j N_j + pb_{r,j} - pb_{s,j} + DCB^\Phi_r - DCB^\Phi_s + m_j + e_j \qquad (2)$$
With r, s, j and i the receiver, satellite, frequency and epoch indices, and where:
$P^s_{r,j}(i)$          Pseudo-range measurements, in meter,
$\Phi^s_{r,j}(i)$          carrier-phase measurements, in meter,
$\rho^s_r(i)$          the geometric distance between satellite and receiver antennas, in meters,
c          the speed of light, in meters per second,
$dt_r$ and $dt^s$          receiver and satellite clock errors, respectively, in seconds,
$T^s_r$          the neutral troposphere delay, in meters,
$I^s_{r,j,P}$ and $I^s_{r,j,\Phi}$          the ionosphere delay of pseudo range and carrier phase observations, in meters,
$N_j$          carrier-phase integer ambiguities, in cycles,
$\lambda_j$          carrier-phase wave length, in meters,
$DCB^p_{r,}$ and $DCB^p_s$    receiver and satellite pseudo-range hardware delays, respectively in metric units,
$DCB^\Phi_r$ and $DCB^\Phi_s$    receiver and satellite carrier-phase hardware delays, respectively, in metric units,
Mj          Pseudo-range multipath on, in meters,
Ej          Other un-modeled errors of pseudo-range measurements, in meters,
pbr,i and pbs,i      receiver and satellite carrier-phase initial phase bias, respectively, in metric units,
mj          carrier-phase multipath, in meters and
ej          Other un-modeled errors of carrier-phase measurements, in meters.
Here, we consider a measurement scenario that one GPS receiver tracks dual frequency code and phase data from a total of m
satellites over t epochs, thereby implying r = 1, s = 1, ….. m, j = 1, 2 and i = 1, ….., t.
Firstly, the code read the Rinex files and extract the pseudo range and carrier phase observations which are the range distances
between the receivers and satellites measured using $L_1$ and $L_2$ frequencies. The "geometry-free" linear combination of GPS
observations is used to derive the observable. The geometric range, clock-offsets and tropospheric delay are frequency
independent and can be eliminated using this combination. The "geometry-free" linear combinations for pseudo range and
carrier phase observations are given as (Al-Fanek 2013):
$$P_4 = P^s_{r,1}(i) - P^s_{r,2}(i) = I^S_{r,1,p} - I^S_{r,2,p} + DCB^p_r + DCB^p_s + E_{12} \qquad (3)$$
$$\Phi_4 = \Phi^s_{r,1}(i) - \Phi^s_{r,2}(i) = I^S_{r,2,\Phi} - I^S_{r,1,\Phi} + \lambda_1 N_1 - \lambda_2 N_2 + DCB^\Phi_r + DCB^\Phi_s + e_{12} \qquad (4)$$
$E_{12} = \sqrt{(E_1)^2 + (E_2)^2}$     is the combination of multipath and measurement noise on $P^s_{r,1}(i)$ and $P^s_{r,2}(i)$ (m), and
$e_{12} = \sqrt{(e_1)^2 + (e_2)^2}$     is the combination of multipath and measurement noise on $\Phi^s_{r,1}(i)$ and $\Phi^s_{r,2}(i)$ (m).
To reduce the multipath and noise level in the pseudo range observables, the carrier phase measurements are used to compute
a more precise relative smoothed range. Although the carrier-phase observables are more precise than the code derived, they
are ambiguous due to the presence of integer phase ambiguities in the carrier phase measurements. To take advantage of the
low-noise carrier phase derived and unambiguous nature of the pseudo range, both measurements are combined to collect the
best of both observations.
Smoothed $P_{4,sm}$ observations can be expressed as follows (Jin et al. 2012):
$$P_{4,sm} = \omega_t P_4(t) + (1 - \omega_t)P_{4,prd}(t) \qquad (t > 1) \qquad (5)$$
where t stands for the epoch number, $\omega_t$ is the weight factor related with epoch t, and
$$P_{4,prd}(t) = P_{4,sm}(t - 1) + [L_4(t) - L_4(t - 1)] \qquad (t > 1) \qquad (6)$$
when t is equal to 1, which means the first epoch of one observation arc, $P_{4,sm}$ is equal to $P_4$.

**3. Spherical Harmonic Model**
To determine the receiver DCB, there are two different methods. The first one is to calibrate the receiver device and obtain the
DCB directly. This method calculates the DCB of the receiver device ignoring that from the antenna cabling used during
observation (Hansen, 2002). The second method calculates the receiver DCB as a part of GPS signal time delay which is
independent on type of antenna. MSDCBE code works as the second methods (figure1).
The ionosphere delay can be expressed as follows (Abid et al. 2016):
$$d_{ion} = \frac{40.3}{f^2} STEC \qquad (7)$$
Where f stands for the frequency of the carrier and Slant Total Electron Content (STEC) is the total electron content along the
path of the signal. The observation equation can be formed by Substituting (7) into (3), and replacing $P_4$ by
smoothed $P_{4,sm}$, we get (Abid et al. 2016):
$$P_{4,sm} = 40.3\left(\frac{1}{f_1^2} - \frac{1}{f_2^2}\right)STEC + c * DCB_r + c * DCB_s \qquad (8)$$
Where: c is the speed of light and $DCB_r$ and $DCB_s$ are differential code bias for receiver and satellites in seconds.
STEC can be translated into Vertical Total Electron Content (VTEC) using the modified single-layer model (MSLM) (Haines
1985, Jin et al. 2012):
VTEC = MF(z)STEC                                                                 (9)
$MF = \cos\left(arcsin\left(\frac{R}{R+H}sin(\alpha z)\right)\right)$                                                           (10)
Where:
MF    is the mapping function,
z      is the satellite elevation angle,
R      is the radius of the Earth=6371 km and
H      is the attitude of the ionosphere thin shell (assumed as used by CODE=506.7 km), $\alpha$=0.9782 (Jin et al. 2012).
To estimate the satellite and receiver HDs, the current study applies a model based on spherical harmonic function to calculate
them using zero-difference observations. The used model is expressed as follows (Schaer 1999, Li et al. 2015, Elghazouly et
al, 2019):
$VTEC(\beta,s) = \sum_{n=0}^{N}\sum_{m=0}^{n} P_n^m(sin(\beta))(A_n^m cos(m\lambda) + B_n^m sin(m\lambda))$                           (11)
Where:
$\beta$   is the geocentric latitude of IPPs (Ionosphere Peirce Point),
s    is the solar fixed longitude of IPPs,
N   is the degree of the spherical function,
M   is the order of spherical harmonic function,
$P_{mn}$ is regularization Legendre series and
$A_{mn}$ and $B_{mn}$ are the estimated spherical harmonics coefficients.
By substituting eq (10) and eq (11) into eq (9) we get:
$$\sum_{n=0}^{N}\sum_{m=0}^{n} P_n^m(sin(\beta))(A_n^m cos(m\lambda) + B_n^m sin(m\lambda))$$
$$= \cos\left(arcsin\left(\frac{R}{R+H}sin(\alpha z)\right)\right)\left[-\frac{f_1^2 f_2^2}{40.3(f_1^2 - f_2^2)}(P_{4,sm} - c*DCB_r - c*DCB_s)\right]$$   (12)
Only one GPS station has more than 20,000 observations per a day. When applying equation (12) using stations observation
data, there are number of equations much more than the number of unknown coefficients. These coefficients were determined
using weighted least square method. general form of weighted least square function can be expressed as (Ghilani and Wolf,
137 2012):

$X = (A^T PA)^{-1} A^T PL$                                                             (13)
Where:
X        is the unknown parameters vector namely, $A_n^m, B_n^m, DCB_r$ and $DCB_s$ ,
A        is the coefficient (design) matrix (coefficients of $A_n^m, B_n^m, DCB_r$ and $DCB_s$),
L        is the observation vector (values of $P_{4,sm}$ ) and
P        is the weight matrix.
As known, the quality of observations is affected by satellite elevation angle, each observation has a weight value depend on
its satellite elevation angle. The weight value can be computed from the following equations (14, 15 and 16) (Luo X., 2013):
$w = \frac{\sigma_0^2}{\sigma^2}$                                                                  (14)
$\sigma^2 = \left[0.05 + \frac{0.02}{sin(z)^2}\right]^2$                                                           (15)
$\sigma_0^2 = (c + d)^2$                                                               (16)
Where:
c & d     are two constants equal to 5 and 2 cm, respectively,

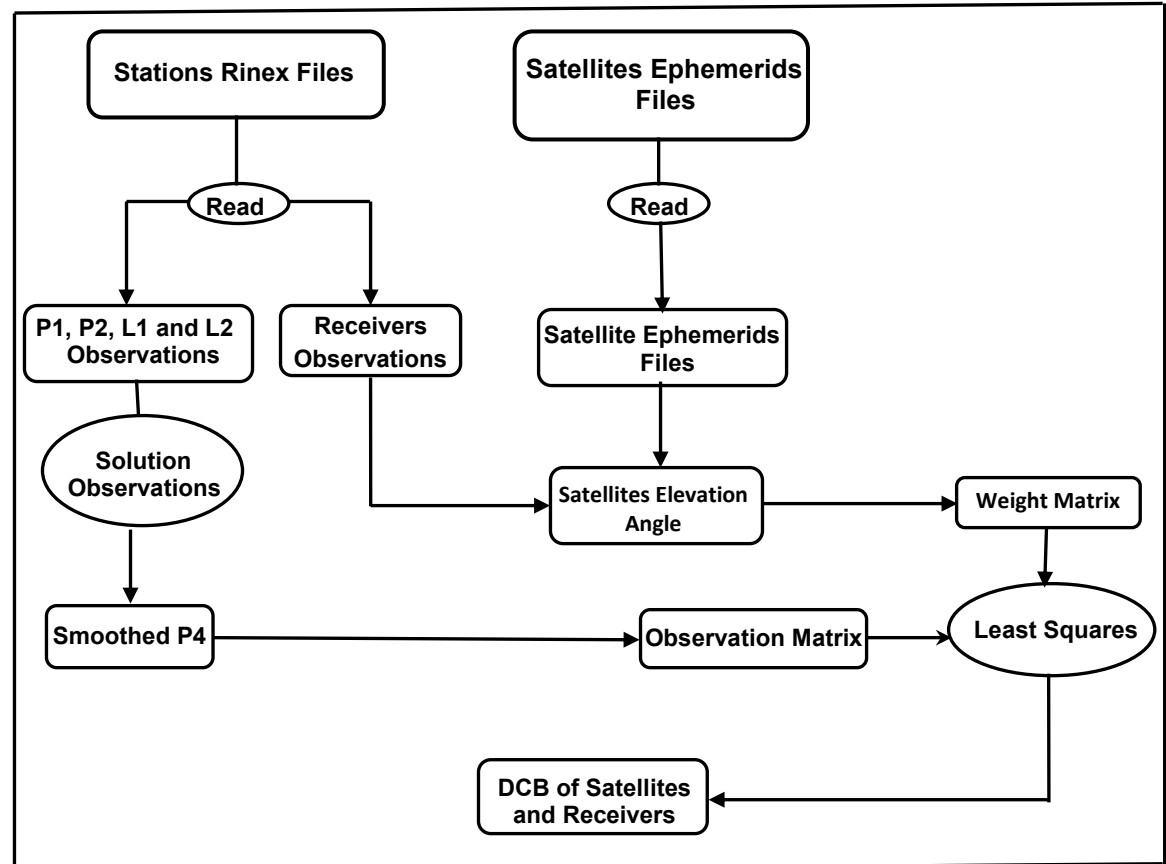

**Figure 1** Flow chart shows how the code works

## 1. Mathematical Model Evaluation

The MSDCBE software was written in MATLAB (version 2016a). The first input is GPS observations in Receiver Independent Exchange (RINEX) format according to the selected stations (figure 2) downloaded from (ftp://garner.ucsd.edu/rinex) and precise ephemerides (SP3) files of test days downloaded from (http://www.GPScalendar.com/index.html?year=2010). In addition, IONosphere Map EXchange Format (IONEX) files of IGS, CODE and JPL are downloaded - as a threshold values - from (ftp://cddis.gsfc.nasa.gov/GPS/products/ionex/).

In the present contribution, to evaluate the performance of the developed model, numerical case-studies were performed. The main goals of the numerical case-studies are to investigate three issues:

**First issue** is to investigate the effect of applying weighted least square instead of least square on satellites and GPS receiver DCBs, and this is done by comparing results from MSDCBE which applying weighted least square with the published results of M_DCB by Jin et al. (2012), and with those of IAAC.

BOGO, BRUS, GOPE, GRAS, ONSA, POTS, PTBB, SOFI and WTZA IGS Stations data from 1 to 31 January 2010 were applied as it was the same network used by Jin et al. (2012).

**Second issue** is to investigate the correlation between size (number of receivers) of the GPS network and estimated DCBs for satellite and GPS receiver, and this is done by comparing DCB values of three stations namely, GOPE, GRAS and ONSA estimated from a network consists of 3 GPS receiver and a network consists of 9 GPS receiver.

This study was applied using IGS Stations data from 1 to 5 January 2010 of six stations namely, BOGO, BRUS, GOPE, GRAS, ONSA, POTS, PTBB, SOFI and WTZA.

**Third issue** is to investigate the congruence of DCBs estimated from absolute and relative methods with other IAAC, and this is done by comparing results from MSDCBE with the published results of ZDDCBE by Sedeek et al. (2017).

This study was applied using data from 1 to 5 January 2010 of six stations namely, GOPE, GRAS, ONSA, MADR, PTBB, and SOFI which was the same network used by Jin et al. (2012) and Sedeek et al. (2017).

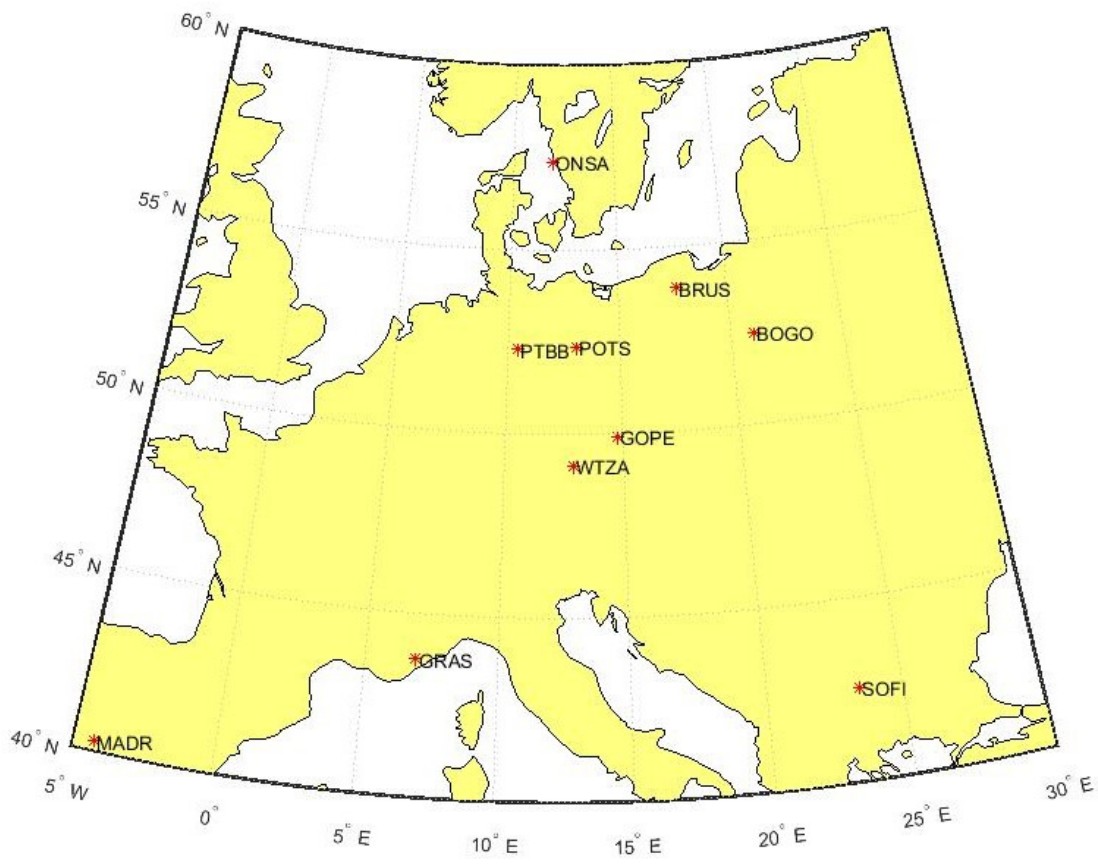

**Figure 2** IGS Stations locations

### *Comparison of multi-station test results from MSDCBE and M_DCB*

The first evaluation made by this paper is the evaluation of weight function. MSDCBE used a weight function depending on
the satellite elevation angle as mentioned before. Table 1 shows the differences and RMS between satellites and receivers
estimated from 1 to 31 January 2010 using multiple GPS stations of both MSDCBE (weighted) and M_DCB (unweighted).
**Table 1** the differences and RMS between satellites and receivers estimated from 1 to 31 January 2010 using multiple GPS
stations (MSDCBE and M_DCB minus CODE).

| satellite | MSDCBE | | M_DCB | | satellite | MSDCBE | | M_DCB | |
|---|---|---|---|---|---|---|---|---|---|
| | differences(ns) | RMS | differences(ns) | RMS | | differences(ns) | RMS | differences(ns) | RMS |
| G1 | 0.228 | 0.250 | 0.746 | 0.251 | G17 | 0.087 | 0.125 | 0.038 | 0.138 |
| G2 | 0.121 | 0.091 | -0.073 | 0.087 | G18 | -0.136 | 0.113 | -0.044 | 0.100 |
| G3 | 0.004 | 0.078 | 0.194 | 0.066 | G19 | 0.236 | 0.095 | 0.381 | 0.066 |
| G4 | 0.169 | 0.092 | 0.003 | 0.123 | G20 | 0.096 | 0.096 | 0.004 | 0.073 |
| G5 | -0.082 | 0.106 | -0.236 | 0.111 | G21 | -0.208 | 0.109 | -0.121 | 0.088 |
| G6 | -0.059 | 0.066 | 0.169 | 0.061 | G22 | -0.188 | 0.091 | 0.050 | 0.109 |
| G7 | -0.015 | 0.084 | -0.233 | 0.085 | G23 | 0.210 | 0.082 | 0.052 | 0.053 |
| G8 | -0.094 | 0.085 | -0.271 | 0.085 | G24 | -0.168 | 0.086 | -0.221 | 0.076 |
| G9 | 0.011 | 0.074 | 0.038 | 0.088 | G25 | -0.091 | 0.122 | -0.220 | 0.085 |
| G10 | -0.068 | 0.088 | -0.343 | 0.095 | G26 | -0.302 | 0.089 | -0.020 | 0.092 |
| G11 | 0.211 | 0.090 | 0.202 | 0.063 | G27 | 0.078 | 0.062 | 0.060 | 0.088 |
| G12 | 0.029 | 0.059 | 0.049 | 0.051 | G28 | -0.177 | 0.080 | -0.340 | 0.107 |
| G13 | 0.296 | 0.080 | 0.140 | 0.062 | G29 | -0.195 | 0.128 | -0.277 | 0.091 |
| G14 | -0.058 | 0.124 | 0.150 | 0.126 | G30 | 0.057 | 0.077 | 0.020 | 0.074 |
| G15 | -0.055 | 0.101 | -0.164 | 0.117 | G31 | 0.018 | 0.099 | 0.057 | 0.138 |
| G16 | -0.057 | 0.069 | 0.096 | 0.084 | G32 | 0.102 | 0.070 | 0.115 | 0.077 |
| BOGO | 0.139 | 0.077 | 0.065 | 0.080 | POTS | 0.120 | 0.073 | 0.237 | 0.094 |
| BRUS | 0.121 | 0.120 | 0.309 | 0.111 | PTBB | 0.083 | 0.082 | 0.201 | 0.095 |
| GOPE | 0.150 | 0.069 | 0.142 | 0.068 | SOFI | -0.045 | 0.119 | 0.081 | 0.113 |
| GRAS | 0.085 | 0.125 | 0.370 | 0.131 | WTZA | 0.137 | 0.078 | 0.270 | 0.083 |
| ONSA | 0.140 | 0.093 | 0.178 | 0.103 | | | | | |

From the table one can see that the differences of MSDCBE estimated satellites DCBs are less than 0.302 ns and the RMS of
all satellites DCBs differences are less than 0.128 except G1 whose RMS = 0.250. The maximum difference of MSDCBE
estimated receivers DCBs is 0.150 ns of receiver GOPE and the minimum is 0.045 ns of receiver SOFI (Figure 3). The
maximum RMS of MSDCBE estimated receivers DCBs is 0.125. On the other side, M_DCB results show that Receiver DCB

biases are slightly larger than those for satellites, but most of them are less than 0.4 ns except G1 whose DCB bias reaches 0.746 ns. The RMS of all differences is lower than 0.3 ns (Jin et al. 2012). Figure 4 shows the mean differences between receiver DCB values estimated by MSDCBE and those released by CODE, IGS, and JPL combined from 1-31 Jan 2010. The figure shows that the results of MSDCBE are mostly close to those of CODE than IGS and JPL. By comparing the figure 4 with the corresponding chart published by Jin et al. (2012), it is clearly appeared that all differences between MSDCBE receivers' DCBs results and between CODE, IGS and JPL are less than those from M_DCB except station GOPE almost equal.

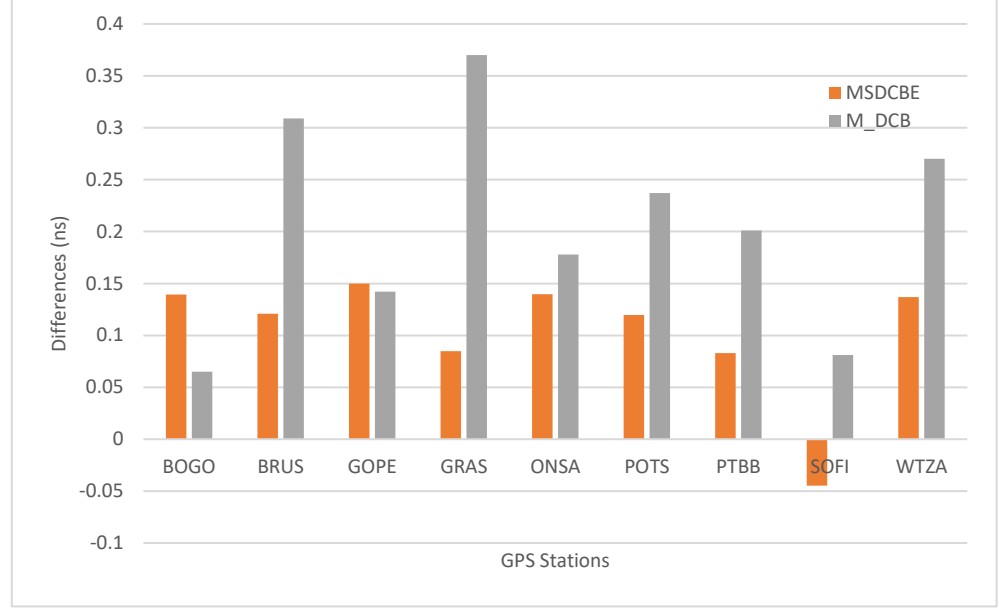

**Figure 3** Mean difference between the receiver DCB values of CODE and the computed values by each of MSDCBE and M_DCB estimated from (1-31) Jan 2010.

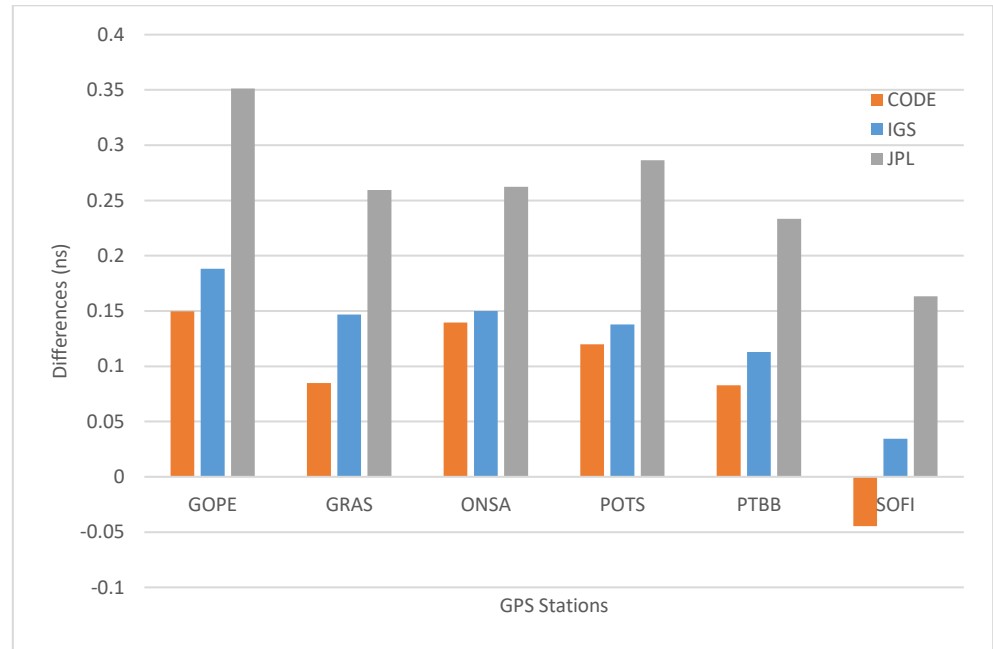

**Figure 4** Mean differences between receiver DCB values estimated by MSDCBE and those released by CODE, JPL, and IGS combined from 1-31 Jan 2010.

### *Effect of network size factor on DCB estimation*

By using multi station DCBs estimation, the number of stations used will appear as a factor influences DCBs estimation. This test was done by comparing DCBs computed by MSDCBE of a network of three receivers namely GOPE, GRAS, ONSA and DCBs of the same receivers but this time as a part of a network of nine receivers namely BOGO, BRUS, GOPE, GRAS, ONSA, PTBB, SOFI and WTZA. Figure 5 shows these results which demonstrate that using nine receivers gives more accurate DCBs. Also, the satellites DCBs differences (figure 6) almost improved but not like receivers DCBs, because satellites DCBs are small values compared with those of receivers.

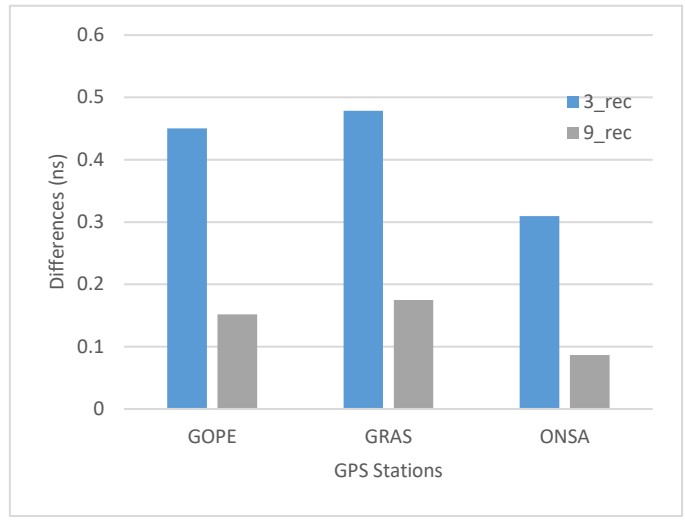

**Figure 5** Mean difference between the receiver DCB values of IGS and the computed values by MSDCBE estimated from (1-5) Jan 2010.

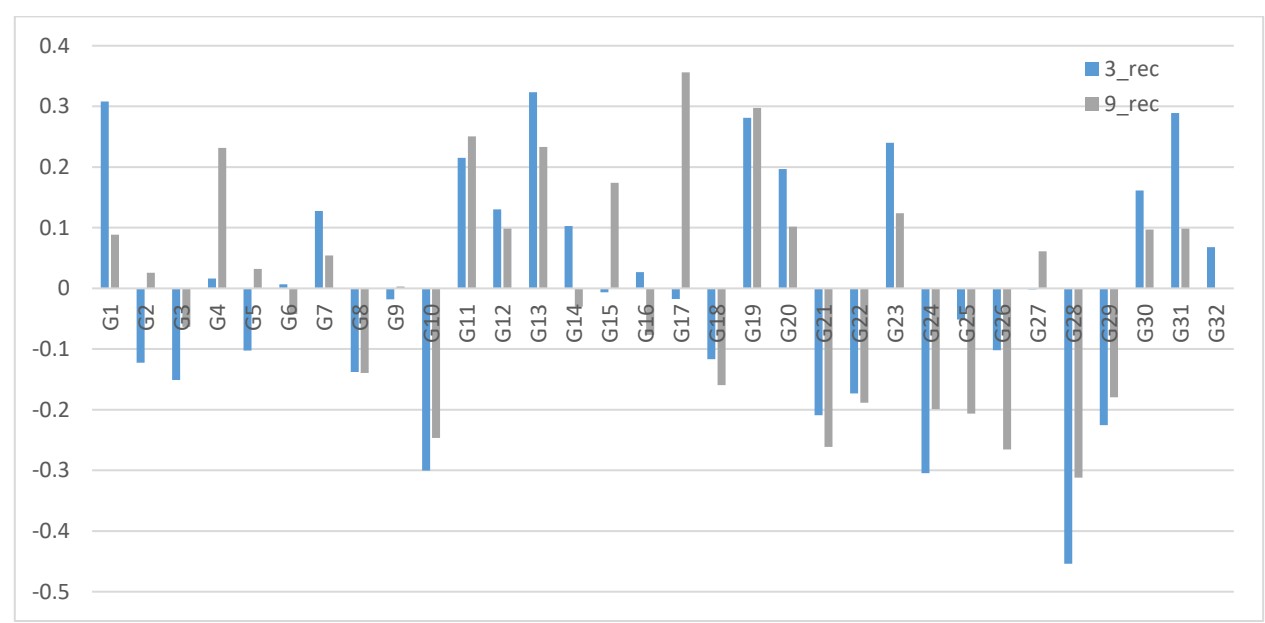

**Figure 6** Mean difference between the satellites DCB values of IGS and the computed values by MSDCBE estimated from (1-5) Jan 2010

## *Comparison of multi-station from MSDCBE and single station from ZDDCBE and M_DCB test results*

In this section the performance of multi station network against single station DCB estimation will be evaluated. Table 2 shows the mean deference between the receiver DCB values computed by IGS and the computed values by each of M_DCB, ZDDCBE and MSDCBE estimated from 1-5 Jan 2010. Figure 7 shows these results graphically and figure 8 shows the mean differences computed from M_DCB, ZDDCBE and MSDCBE for GPS satellites. The results show a significant difference between multi station network against single station DCB estimation. The maximum difference between receiver DCB estimation using IGS and MSDCBE is 0.1477 ns of MADR station, but it is 1.1866 ns and 0.7982 ns for M_DCB and ZDDCBE respectively.

**Table 2** Mean deference between the receiver DCB values computed by IGS and the computed values by using single station M_DCB, ZDDCBE and multi-station MSDCBE estimated from 1-5 Jan 2010.

| IGS St. | Model | DCB diff. (ns) | IGS St. | Model | DCB diff. (ns) |
|---------|-------|----------------|---------|-------|----------------|
| **GOPE** | M_DCB | 0.3847 | **ONSA** | M_DCB | 1.1866 |
| | ZDDCBE | 0.1724 | | ZDDCBE | 0.7982 |
| | MSDCBE | 0.004 | | MSDCBE | -0.0310 |
| **GRAS** | M_DCB | 0.3379 | **PTBB** | M_DCB | 0.6692 |
| | ZDDCBE | 0.1466 | | ZDDCBE | 0.3550 |
| | MSDCBE | 0.066 | | MSDCBE | -0.0578 |
| **MADR** | M_DCB | 0.3078 | **SOFI** | M_DCB | 0.6916 |
| | ZDDCBE | 0.3468 | | ZDDCBE | 0.4650 |
| | MSDCBE | 0.1477 | | MSDCBE | -0.0149 |

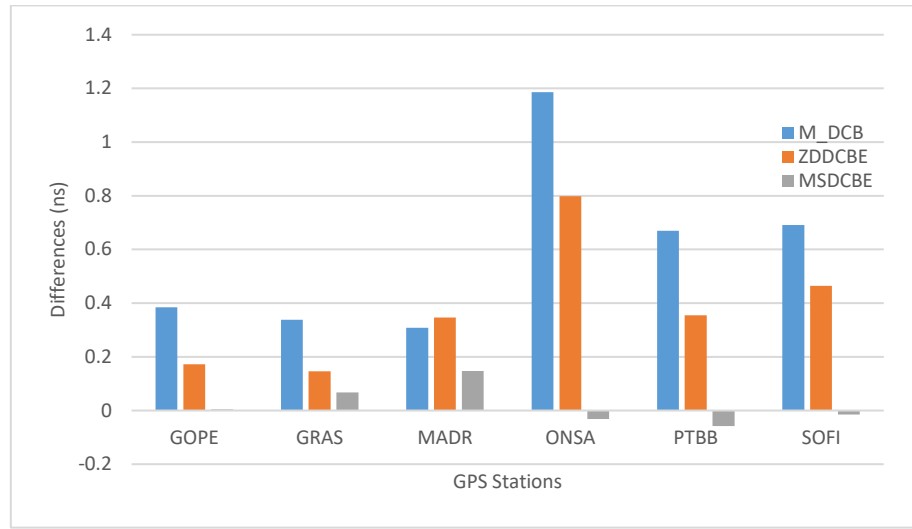

223

**Figure 7** Mean difference between the receiver DCB values of IGS and the computed values by each of M_DCB, ZDDCBE and MSDCBE estimated from (1-5) Jan 2010

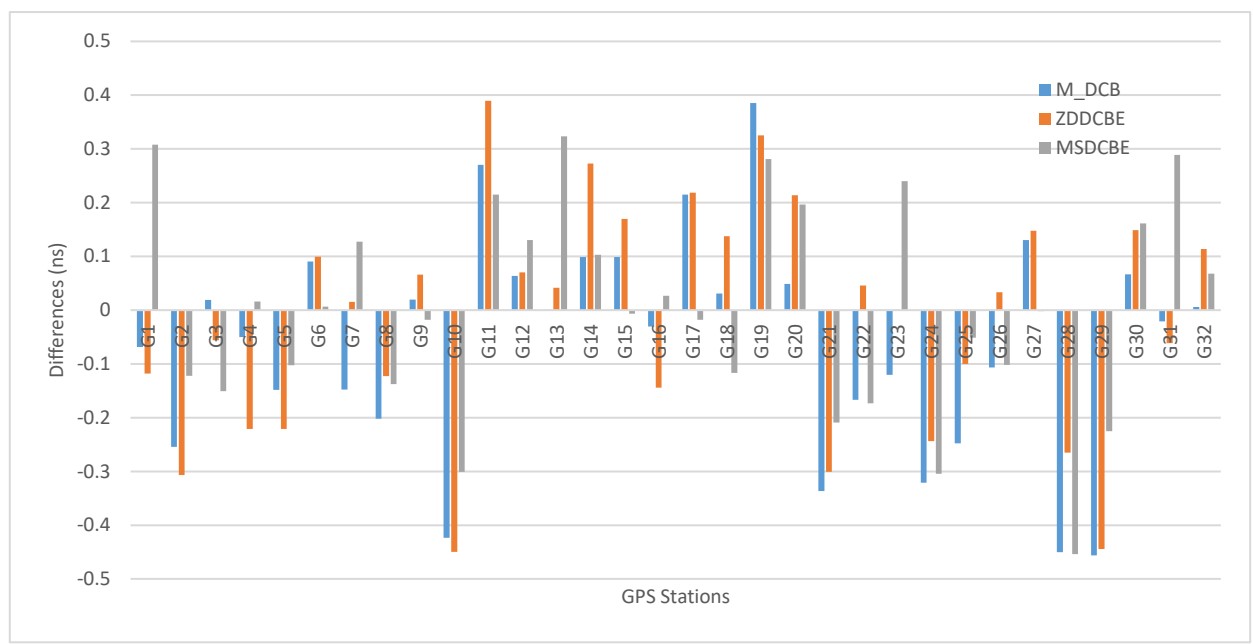

226

**Figure 8** Mean difference between the satellites DCB values of IGS and the computed values by M_DCB, ZDDCBE and MSDCBE estimated from (1-5) Jan 2010

**Conclusions**

The current study proposes a new MATLAB code called MSDCBE able to calculate DCBs of GPS satellites and receivers. This code was compared with two other codes and evaluated using IAAC data and from all the above, we can conclude that:

1.  The estimated DCBs results also affected and improved by using weight function according to satellite elevation angle observations. In addition, results show a good agreement with IGS, CODE and JPL results than using multi station estimation DCB without weight function.
2.  When using multi station DCB estimation, number of input stations influences in DCB results. However, it is recommended to enlarge the size of used network, but it needs high computer requirements and much more analysis time (only one station have more than 20,000 observation per a day).
3.  The most effective factor in DCBs estimation is using multi station network instead of single station that appeared from results which improved from 1.1866 ns and 0.7982 ns maximum DCB mean differences for M_DCB and ZDDCBE single station analysis to 0.1477 ns for MSDCBE. So, using multi station network DCB estimation- if available- is strongly recommended.

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
