# Peer review of "Estimating Satellite and Receiver Differential Code Bias Using Relative GPS Network"

_Annales Geophysicae, 2018_

## Referee Comment (RC1) · Anonymous Referee #1 · 9 Dec 2018

The author proposed three conclusions:

1. the DCB results can be improved by using weight function according elevation;

2. more stations used in DCB estimation can improve the results precision;

3. it is better using multi station network than single station.

However, the conclusions above are widely known across the community and thus not new.

Furthermore, I have the following specific remarks:

-The experiment data should be more in spatial and temporal resolution, which means the span of the data should be longer and the number of stations should be more.

-Many abbreviations should be specified at the beginning of the manuscript where they first appear.

-The author should clearly point out the form of weighted function in the manuscript.

---

## Author Comment (AC2) · 22 Feb 2019

I am grateful to this journal and would like to thank editor very much for letting my manuscript under discussion in this valuable journal. But i need to withdraw my manuscript due to some personal reasons, please.

---

## Referee Comment (RC2) · Anonymous Referee #2 · 8 May 2019

In the article the authors suggested a technique for "Estimating Satellite and Receiver Differential Code Bias". While the problem of DCB estimation is important enough it is difficult to find what new was done in the article. The authors used well-known approach based on spherical harmonics (SH), which was suggested by S. Schaer et al. (1998). Similar approach based on SH was used for M_DCB: MATLAB code by Jin et al. (2012). Recently a lot of article have been published containing new results on DCB and DCB estimation, for example applying convolution algorithm (Q. Li et al., 2018 ), applying SH along with trigonometric series (Z. Li et al., 2015), applying spherical cap harmonic for regional modeling (Liu et al., 2010), using combination of Minimum scalloping/Least squares/ Zero TEC method (Rideout & Coster, 2006), as well as indicating problems and solution for Compass/Beidou DCB (Z. Li et al.,

2012), new results on strong annual DCB variations (Mylnikova et al., 2015), grounding influence on DCB (Choi and Lee, 2018), plasmasphere influence on DCB estimation (Themens et al., 2015) etc. Submitted article by Elghazouly et al. does not use the background.

Another issue is that in Europe there are at least several hundred stations. Correct analysis (see "second issue" in the article) should contain results (for several stations) for densest network, less dense, . . . one station.

There are also a general problem: nobody knows the real DCB. That make me doubting about 3rd conclusion "The most effective factor in DCBs estimation is using multi station network instead of single station that appeared from results which improved from 1.1866 ns and 0.7982 ns maximum DCB mean differences for M_DCB and ZDD-CBE single station analysis to 0.1477 ns for MSDCBE. So, using multi station network DCB estimation- if available- is strongly recommended". The results only say that used techniques are similar. So, that requires testing the technique based on the modeling results.

Such requirements are for article submitted to Annales geophysicae. It seems that the authors would like to publish "software article" ("The current study proposes a new MATLAB code"), so I would recommend to look for "software journal" (like, for example, The Journal of Open Source Software).

Minor comments:

1) While the article contains some interesting results the poor organization of the article make it difficult to understand and make sure that they are correct.

2) There are a lot of formulas in the article but actually only 12-16 are used

3) There are different errors. "By substituting eq (11) and eq (13) into eq (10) we get". Actually (8), (9) and (10) into (11). "following equations (14, 15 and 18)" – there is no (18).

References:

B.-K.Choi, S.J. Lee, The influence of grounding on GPS receiver differential code biases, Advances in Space Research, V. 62, Issue 2, 2018, P. 457-463, https://doi.org/10.1016/j.asr.2018.04.033.

Jin, R., Jin, S., and Feng, G.: M_DCB: MATLAB code for estimating GPS satellite and receiver differential code biases, GPS 243 Solution 16:541–548, 2012.

Z.Li, Y. Yuan, H.Li, J.Ou, X. Huo, Two-step method for the determination of the differential code biases of COMPASS satellites // J Geod (2012) 86:1059–1076. DOI 10.1007/s00190-012-0565-4.

Z. Li, Y. Yuan, N. Wang, M. Hernandez-Pajares, X. Huo. SHPTS: towards a new method for generating precise global ionospheric TEC map based on spherical harmonic and generalized trigonometric series functions. J Geod (2015) 89: 331. https://doi.org/10.1007/s00190-014-0778-9

Q. Li, G. Ma, W. Lu, Q. Wan, J. Fan, X. Wang, J.Li, C.Li, A method of estimating GPS instrumental biases with a convolution algorithm, Advances in Space Research, V. 61, Issue 6, 2018, P. 1387-1397, https://doi.org/10.1016/j.asr.2017.11.034.

J. Liu, R. Chen, H. Kuusniemi, Z. Wang, H. Zhang, and J. Yang (2010), A preliminary study on mapping the regional ionospheric TEC using a spherical cap harmonic model in high latitudes and the Arctic Region, J. Glob. Pos. Syst. 9, 1, 22-32, DOI: 10.5081/jgps.9.1.22.

A.A. Mylnikova, Yu.V. Yasyukevich, V.E. Kunitsyn, A.M. Padokhin, Variability of GPS/GLONASS differential code biases, Results in Physics, V. 5, 2015, P. 9-10, https://doi.org/10.1016/j.rinp.2014.11.002.

Rideout, W. & Coster, A. Automated GPS processing for global total electron content data. GPS Solut (2006) 10: 219. https://doi.org/10.1007/s10291-006-0029-5.

D. R.Themens, P.T. Jayachandran, R.B. Langley ( 2015), The nature of GPS differential receiver bias variability: An examination in the polar cap region, J. Geophys. Res. Space Physics, 120, 8155– 8175, doi:10.1002/2015JA021639.

S.Schaer, G.Beutler, M.Rothacher MAPPING AND PREDICTING THE IONOSPHERE // Proceedings of the IGS AC Workshop, Darmstadt, Germany, February 9–11, 1998.

---

## Author Comment (AC3) · 27 May 2019

the file of the response uploaded in pdf format.

Please also note the supplement to this comment:
https://www.ann-geophys-discuss.net/angeo-2018-120/angeo-2018-120-AC3-supplement.pdf
* * *

---

## Author Comment (AC1)

**Reply to the review of the Anonymous Referee #1:**

The Authors are grateful to the editor and would like to thank the Referee #1 very much for his important comments that helped us to improve the original manuscript. We have responded to all comments. Details of our responses to each comment are shown below:-

| NO. | Referee's Comments | Authors Responses |
|---|---|---|
| 1 | 1. the DCB results can be improved by using weight function according elevation;
2. more stations used in DCB estimation can improve the results precision;
3. it is better using multi station network than single station.
 However, the conclusions above are widely known across the community and thus not new. | The main objective of our paper is to introduce our new code for estimating satellites and receivers DCB values and check its validity to produce precise DCBs in different cases. So, when we say that DCB results can be improved by using weight function according elevation or more stations, we want to conclude that our code gives more precise results compared with other codes of other researchers. It can be clearly appearing in the first two lines of the Conclusion section (line number 221 and 222). |
| 2 | - The experiment data should be more in spatial and temporal resolution, which means the span of the data should be longer and the number of stations should be more. | The validation of the code was made by comparing with other researchers' code. To compare our results with other researchers (Jin et al, GPS Solution 16:541–548, 2012) and (Sedeek te al., Arab J Geosci, DOI 10.1007/s12517-017-2835-1, 2017) results we should use the same receivers number at the same days. |
| 3 | Many abbreviations should be specified at the beginning of the manuscript where they first appear. | The manuscript was revised and we note the following abbreviations were missed:
**CODE** Center for Orbit Determination in Europe
**IGS**    International GNSS Service
**UHF**    Ultra High Frequency
**STEC**    Slant Total Electron Content |
| 4 | The author should clearly point out the form of weighted function in the manuscript. | The form of the weight function introduced in lines number 138, 139 and 140, but it was really missed the values of c and d which are equal to 5 and 2 cm. In addition, z in equation (15) is the satellite elevation angle that was defined in line number (111). |

---

## Author Response (AR1)

Title: Estimating Satellite and Receiver Differential Code Bias Using Relative GPS Network Advances in Space Research

Dear Editor,

We have modified the manuscript and detailed corrections are listed below point by point:

1. Many abbreviations should be specified at the beginning of the manuscript where they first appear.
   following abbreviations were added:
   CODE: Center for Orbit Determination in Europe
   IGS:    International GNSS Service
   STEC:  Slant Total Electron Content
2. The author should clearly point out the form of weighted function in the manuscript
   The form of the weight function introduced in lines number 146, 147 and 148, but it was really missed the values of c and d which are equal to 5 and 2 cm. In addition, z in equation (15) is the satellite elevation angle that was defined in line number (117).
3. There are different errors. "By substituting eq (11) and eq (13) into eq (10) we get". Actually (8), (9) and (10) into (11). "following equations (14, 15 and 18)" – there is no (18).
   The mentioned parts were corrected.

The manuscript has been resubmitted to your journal. We look forward to your positive response.
Sincerely,

Alaa Elghazouly
Associate Lecturer
Civil Engineering Department
Faculty of Engineering, Menoufia University, Egypt

Mob:      +2-01001721508
E-mail:    alaa_elghazouly@sh-eng.menofia.edu.eg

[revised manuscript text omitted]

---

## Referee Report (RR1)

The manuscript presents results of GPS satellite and receiver differential code bias using multi-stations. For this study authors have used the weighted least square to estimate satellite and receiver DCBs. However, this software is similar to the software of Jin et al. (2012), which is only to use the algorithm of weighted least square. Generally, the paper is not sufficiently innovative.

Other comments as follow:

1. The model of spherical harmonic function is key to calculate the DCBs. However, the order of spherical harmonic function is very important. How many is the order in this paper? The authors should express clearly in the article.

2. What is the time required to calculate the DCBs of multi stations? For example, 20 stations and 30 stations.

3. In the section of experiment, it is important to select more stations for comparative analysis.

---

## Author Response (AR2)

**Reply to the review of the Anonymous Referee #3:**

The Authors are grateful to the editor and would like to thank the Referee #3 very much for his important comments that helped us to improve the original manuscript. We have responded to all comments. Details of our responses to each comment are shown below: -

| NO. | Referee's Comments | Authors Responses |
|-----|--------------------|--------------------|
| 1 | The authors should express the difference between this software and software of Jin et al (2012) in detail. | As shown in the introduction section (L 50-54), in this study we introduce a mathematical model estimating satellites & receiver DCBs for a GPS network based on Spherical Harmonic Function like M_DCB software. But, the DCB and ionosphere coefficients can be estimated from GPS dual-frequency observations by the Weighted Least Squares (WLS) method. Weights were produced from the satellites elevation angle. Also we can estimate DCB for any type of receiver (codeless tracking, Cross Correlation, and Non-Cross Correlation types) the other software of Jin et al (2012) calculate DCB for codeless tracking receiver type only |
| 2 | The model of spherical harmonic function is key to calculate the DCBs. However, the order of spherical harmonic function is very important. How many is the order in this paper? The authors should express clearly in the article. | Fourth order was used as it is recommended for our small areas, and it is mentioned to the used order in the revised paper. |
| 3 | What is the time required to calculate the DCBs of multi stations? For example, 20 stations and 30 stations. | It depends on number of observation from each station of the network and cut off elevation angle. For our solved networks, it takes about 20-30 min, I think it might need about 60-90 min for 20-30 station. |
| 4 | In the section of experiment, it is important to select more stations for comparative analysis. | In the current paper we used a pre-solved networks which had been published by Jin et al. 2012 and others. To evaluate our results. So, we are restricted with the number of stations used by the other papers. But the code is applicable to any number of stations. |

**Reply to the review of the Anonymous Referee #4:**

The Authors are grateful to the editor and would like to thank the Referee #4 very much for his important comments that helped us to improve the original manuscript. We have responded to all comments. Details of our responses to each comment are shown below: -

| NO. | Referee's Comments | Authors Responses |
|---|---|---|
| 1 | The authors have compared DCBs estimated by different methods, such as MSDCBE, M_DCG, and ZDDCBE, and provided by CODE, IGS, and JPL. The difference between them are shown, but the reasons why the difference is large (or small) is not discussed based on the difference of the adopted methods. | As mentioned in the abstract, these differences between MSDCBE and M_DCB come from the added weight function and the processing weighted least square method. In addition, differences between MSDCBE and ZDDCBE come from using network and single station for MSDCBE and ZDDCBE, respectively. |
| 2 | ll. 16, 17: In Abstract, the authors describe "The second factor concerned with estimating DCBs using single GPS Station Precise Point Positioning (PPP) or using GPS network."
However, the results are not shown in this manuscript. | ZDDCBE code used single station to estimate DCB, which we mean by PPP. Compared results of single and multi-stations (ZDDCBE and MSDCBE) shown in the paper. This statement edited in the revised paper. |
| 3 | l. 131, "By substituting eq (10) and eq (11) into eq (9) we get": Equation (8) is also needed. | considered |
| 4 | l. 147: Explain how constants of 0.05 and 0.02 are determined. | Reference added in the revised paper.
For more details please see (Ray and Griffiths, 2008) |
| 5 | l. 150: Explain how constants of 5 and 2 cm are determined. | |
| 6 | l. 150: "c" is used as speed of light. Use another expression. | considered |
| 7 | DCBs estimated in this study are compared with those obtained from CODE or IGS. The authors consider that smaller difference from DCBs estimated by CODE and/or IGS is better. In this paper, the authors show that MSDCBE with a weighting function depending on the satellite elevation angle is better than M_DCB without weighting function.
Is MSDCBE same as M_DCB except only usage of weighting function? The authors concluded that the estimated DCBs are affected an improved by using weighting function according to the satellite elevation angle. To obtain this conclusion, MSDCBE must be same as M_DCB except only usage of weighting function. | As shown in the introduction section (L 50-54), in this study we introduce a mathematical model estimating satellites & receiver DCBs for a GPS network based on Spherical Harmonic Function like M_DCB software. But, the DCB and ionosphere coefficients can be estimated from GPS dual-frequency observations by the Weighted Least Squares (WLS) method. Weights were produced from the satellites elevation angle.
Also our software capable of calculating DCB for any type of receiver (codeless tracking, Cross Correlation, and Non-Cross Correlation types) the other software of Jin et al (2012) calculate DCB for codeless tracking receiver type only |

| 8 | The authors describe "improved" in conclusion, but the correct value of DCBs are unknown. The estimated results becomes close to the those from IGS and CODE by using a weighting function, but it is impossible to conclude "improved". | considered |
|---|---|---|
| 9 | The authors need to compare the method of MSDCBE with that used by IGS and CODE, and discuss the difference among the methods. Especially, the authors need to mention whether the methods adopted by IGS and CODE use a weighting function or not. If they use the same weighting function, the results shown in this manuscript is meaningless. | CODE using the same spherical harmonic function but with different order (15), JPL uses the triangular mesh model to describe the ionosphere while estimating DCB and TEC coefficients, and the IGS values are from the combination of several GNSS analysis centers. |
| 10 | l. 213: What is the difference of ZDDCBE compared to other methods? | ZDDCBE code used single station to estimate DCB. |

---

## Author Response (AR3)

**Reply to the review of the Anonymous Referee #3 :**

The Authors are grateful to the editor and would like to thank the Referee #3  for their review: -

We have modified the manuscript and detailed corrections are listed below point by point:

1. l. 232: "are" could be needed between "results" and "affected".
   - considered

---

## Author Response (AR4)

**Reply to the review of the Topical Editor:**

The Authors are grateful to the Topical Editor for the review: -

We have modified the manuscript and detailed corrections are listed below point by point:

- line 28/29: consider revising the sentence „The global availability of GPS has made it a valuable tool for sensing the Earth' the regional and global ionosphere estimation"
- line 45: replace estimates by estimate
- line 47: As TEC values are dependent on…
- line 72-76: subscript the indices
- line 80: missing space after L1
- Fig. 6: please add some labels.
- line 212: full stop is missing at the end of the sentence.
- line 215: replace deference by difference
- line 228: full stop is missing at the end of the sentence.

All mentioned parts are considered.